# Fast Graph Condensation with Structure-based Neural Tangent Kernel

Submission Id: 2342

## ABSTRACT

The rapid development of Internet technology has given rise to a vast amount of graph-structured data. Graph Neural Networks (GNNs), as an effective method for various graph mining tasks, incurs substantial computational resource costs when dealing with large-scale graph data. A data-centric manner solution is proposed to condense the large graph dataset into a smaller one without sacrificing the predictive performance of GNNs. However, existing efforts condense graph-structured data through a computational intensive bi-level optimization architecture also suffer from massive computation costs. In this paper, we propose reforming the graph condensation problem as a Kernel Ridge Regression (KRR) task instead of iteratively training GNNs in the inner loop of bi-level optimization. More specifically, We propose a novel dataset condensation framework (GC-SNTK) for graph-structured data, where a Structure-based Neural Tangent Kernel (SNTK) is developed to capture the topology of graph and serves as the kernel function in KRR paradigm. Comprehensive experiments demonstrate the effectiveness of our proposed model in accelerating graph condensation while maintaining high prediction performance.

**Relevance Statement:** There is a wide presence of large-scale graph-structured data (e.g., social networks and semantic web) in the Web Internet. However, GNNs, one of the advanced methods for handling graph data, consume significant computational resources when encountering large-scale graphs. Therefore, this paper proposes a method for efficiently condense large-scale graph data without scarifying the expressive ability, and thereby accelerating the execution speed of downstream tasks.

## CCS CONCEPTS

• **Information systems** → **Data mining**; • **Theory of computation** → **Graph algorithms analysis**.

## KEYWORDS

Graph Condensation, Dataset Distillation, Graph Neural Networks, Kernel Ridge Regression, Neural Tangent Kernel

**ACM Reference Format:**

Anonymous Author(s). 2024. Fast Graph Condensation with Structure-based Neural Tangent Kernel. In *The Web Conference 2024 (WWW'24), May 13–17,*

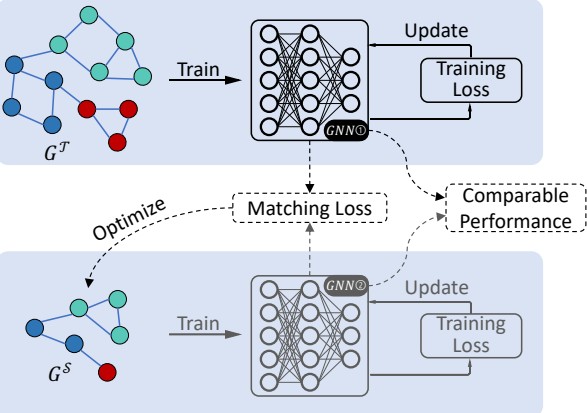

**Figure 1: Graph condensation aims to condense graph data to a smaller but informative version. In general, two GNN classifiers (i.e., GNN① and GNN②) are trained on $G^{\mathcal{T}}$ and $G^{\mathcal{S}}$ simultaneously. Meanwhile, various matching objectives on two GNNs are conducted to synthesize $G^{\mathcal{S}}$.**

*2024, Singapore*. ACM, New York, NY, USA, 10 pages. https://doi.org/10.1145/nnnnnnn.nnnnnnn

## 1 INTRODUCTION

The prevalence of internet technology has accumulated a large amount of graph-structured data, which is widely used in Web applications. For example, social networks [2, 8–10, 48], and semantic networks [23, 53], can be naturally represented as graphs consisting of nodes and edges. Due to the extensive application of graph-structured data, Graph Neural Networks (GNNs) [11, 27, 46], as one of the advanced Deep Neural Networks (DNNs), have gained significant attention for the performance of various graph mining tasks during the past few years. Notably, the remarkable achievement of most existing GNNs largely relies on large-scale datasets [4, 12]. Despite being effective, training GNNs on such large-scale datasets still presents some difficulties, as it usually requires enormous computational resources (e.g., power, memory storage, GPU runtime, etc.) caused by thousands of training iterations, hyper-parameters optimization, and neural architecture search [19, 42].

A data-centric approach to tackle these challenges is to condense the original large-scale graph datasets into synthesized smaller yet information-rich versions [38]. As one of the most advanced paradigms, Dataset Condensation (DC), also known as dataset distillation [51], has achieved remarkable performance to obtain a small-scale synthetic version of the full dataset while preserving models' prediction performance in the image domain. For instance, 10 images distilled from the whole MNIST dataset [21] with 60,000 images are sufficient to achieve a 94% accuracy, reaching 95% performance of the full dataset [38]. Recently, early efforts

explore the potential of dataset condensation on graph-structured data [18, 19]. As shown in Figure 1, the vanilla graph condensation methods [18, 19] can be formulated as a bi-level optimization problem, so as to condense the entire graph into a small graph of a few nodes with corresponding features via various matching objectives between two GNNs models. More specifically, as illustrated in Figure 2(a), the outer synthetic graph optimization (i.e., outer loop) heavily relies on the inner GNN model trained on the synthetic graph (i.e., inner loop) [18, 19].

Despite the aforementioned success, most existing bi-level optimization-based graph condensation methods suffer from intrinsic limitations such as unstable training [29, 31], and massive computation costs [35], leading to an inferior performance on synthesized data generations. Worse still, ensuring condensed data's generalization on various GNN architectures requires multiple parameter initialization [18, 19], leading to complex three nested loops in algorithmic optimization and substantial time consumption.

To tackle the problems, in this paper, we propose to reformulate the graph condensation as a Kernel Ridge Regression (KRR) task in a closed-form solution [6, 30], instead of training GNNs (in the inner loop) with multiple iterations to learn synthetic graph data via bi-level optimization. However, leveraging KRR for graph condensation faces tremendous challenges. The KRR paradigm, which can be treated as a classification task, largely relies on the design of kernel functions to calculate the similarity matrix among instances. The vanilla kernel function, such as dot product and polynomial kernel, might hinder the expressiveness of modeling complex relationships among nodes. Therefore, the first challenge is how to design an appropriate kernel function for optimizing graph condensation in the KRR paradigm. Moreover, unlike images or text data, graph-structured data lies in non-Euclidean space with complex intrinsic dependencies among nodes. In other words, it is imperative to capture graph topological signals in non-linear kernel space. Thus, the second challenge is how to effectively take advantage of topological structure in graphs to enhance the design of kernel function in KRR for graph condensation.

In this paper, we introduce a novel dataset condensation framework for graph-structured data in the node classification task, named **G**raph **C**ondensation with **S**tructure-Based **N**eural **T**angent **K**ernel (**GC-SNTK**). More specifically, a novel graph condensation framework is developed by harnessing the power of the estimated neural kernel into the KRR paradigm, which can be considered as training infinite-width neural networks through infinite steps of SGD. What's more, to capture the topological structure of graphs, a **S**tructure-based **N**eural **T**angent **K**ernel (**SNTK**) is introduced to execute neighborhood aggregation of nodes for generating the high-quality condensed graph. Our contributions can be summarized as follows:

- We propose a principle way based on kernel ridge regression to significantly improve graph condensation efficiency, which can be achieved by replacing the inner GNNs training (i.e., inner loop) via Structure-based Neural Tangent Kernel in the KRR paradigm.
- We propose a novel graph condensation framework GC-SNTK, which can effectively and efficiently synthesize a

smaller graph so that various GNN architectures trained on top of it can maintain comparable generalization performance while being significantly efficient.
- We conduct extensive experiments on various datasets to show the effectiveness and efficiency of the proposed framework for graph condensations. Moreover, our proposed method offers powerful generalization capability across various GNN architectures.

## 2 RELATED WORK

**Dataset Condensation (DC)**. Dataset condensation (DC) [38, 51] aims to condense a large-scale dataset into a small synthetic one, while the model trained on the small synthetic dataset should reach a comparable performance to that trained on the original dataset. [28] introduces the concept of serving the pixels of the training images as hyper-parameters and optimizing them based on gradients. This idea lays the foundation for dataset condensation [38], contributing to the establishment of the basic bi-level optimization framework for DC. Building upon the framework, various criteria for evaluating the performance of models trained on condensed data are proposed, including gradient matching [22, 49, 51], features aligning [37], training trajectory matching [3] and distribution matching [50]. Additionally, Deng et al. introduce the learnable address matrix in condensation [5], which considers the address matrix as part of the condensed dataset. Furthermore, leveraging the support of infinite-width neural network [1, 7, 17, 24], [30] propose a meta-learning approach for image dataset condensation.

**Graph Condensation**. Graph condensation comprises node-level condensation [19] and graph-level condensation [18]. The former focuses on condensing a large graph into a synthetic one with a few nodes, while the latter condenses numerous graphs into a synthetic set containing only a small number of graphs. Among these works, Jin et al. first extend DC to the graph domain and introduce a node-level graph condensation approach [19]. Subsequently, graph condensation is further extended to the graph-level datasets [18], where the discrete graph structure of a synthetic graph is modeled as a probabilistic model. On the other hand, a structure-free graph condensation method [52] is demonstrated as effective. This method condenses a graph into node embedding based on the training trajectory matching. On top of that, Liu et al. propose a node-level graph condensation method built upon the receptive field distribution matching for graph data [25].

## 3 METHODOLOGY

In this section, we start by introducing the bi-level graph condensation model. Next, we provide comprehensive details on the proposed fast graph condensation framework (as shown in Figure 2(b)), and illustrate a structure-based kernel method specifically designed for graph data. Finally, a theoretical analysis of the computational complexity is presented, proving that the proposed GC-SNTK is more efficient compared to previous SOTA method.

### 3.1 Notations and Definitions

In general, a graph can be represented as $G = (\mathcal{V}, \mathcal{E})$, where $\mathcal{V} = \{v_1, v_2, ..., v_{|\mathcal{V}|}\}$ is the set of $|\mathcal{V}|$ nodes and $\mathcal{E}$ is the edge

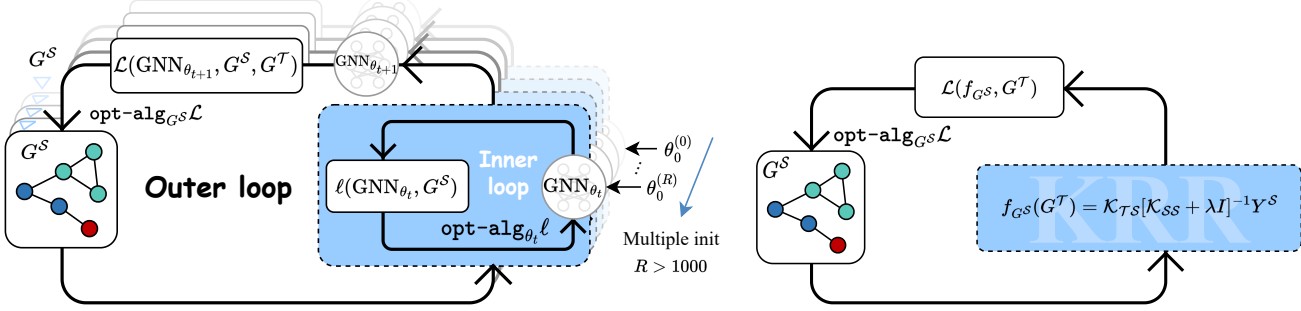

(a) Bi-level graph condensation optimization        (b) Our proposed GC-SNTK

**Figure 2: Bi-level graph condensation optimization (a) and the proposed GC-SNTK (b).** $G^{\mathcal{T}}$ and $G^{\mathcal{S}}$ denote the target and condensed graph data. $\text{GNN}_\theta$ is the graph neural network model with parameter $\theta$. $\mathcal{L}$ and $\ell$ are the loss of the outer and inner loop, respectively. `opt-alg` is the optimization algorithm. The bi-level model entails a inner GNN training loop, a outer $G^{\mathcal{S}}$ optmization loop, and $R$-time initialization. On the contrary, the proposed GC-SNTK only have a single $G^{\mathcal{S}}$ optimization loop.

set. We use $X \in \mathbb{R}^{|\mathcal{V}| \times d}$ to denote the node features matrix, where $d$ is the dimensional size of node features. The structural information of the graph can be represented as a adjacency matrix $A \in \{0, 1\}^{|\mathcal{V}| \times |\mathcal{V}|}$, where $A_{ij} = 1$ indicates that there is a connection between node $v_i$ and $v_j$, and 0 otherwise. Given a target graph dataset $G^{\mathcal{T}} = \{X^{\mathcal{T}}, A^{\mathcal{T}}, Y^{\mathcal{T}}\}$ with $N$ nodes, where $Y^{\mathcal{T}}$ is nodes' labels. The goal of graph condensation is to condense $G^{\mathcal{T}}$ into a graph $G^{\mathcal{S}} = \{X^{\mathcal{S}}, A^{\mathcal{S}}, Y^{\mathcal{S}}\}$ with a significantly smaller nodes number $M$ ($M \ll N$), while the model trained on $G^{\mathcal{S}}$ can achieve comparable performance to that trained on the much larger target graph dataset $G^{\mathcal{T}}$ (as shown in Figure 1).

### 3.2 Bi-level Graph Condensation

As shown in Figure 2(a), existing solutions regards graph condensation as a bi-level optimization problem [18], which can be formulated as follows:

$$\min_{G^{\mathcal{S}}} \mathbb{E}_{\theta_0 \sim P_{\theta_0}} \left[ \sum_{t=0}^{T} \mathcal{L}(\text{GNN}_{\theta_{t+1}}, G^{\mathcal{S}}, G^{\mathcal{T}}) \right]$$

$$s.t. \quad \text{GNN}_{\theta_{t+1}} = \text{opt-alg}_{\theta_t} \left[ \ell(\text{GNN}_{\theta_t}, G^{\mathcal{S}}) \right]. \tag{1}$$

In Equation (1), the outer loop is responsible for optimizing the condensed graph data $G^{\mathcal{S}}$ by minimizing the matching loss $\mathcal{L}$. At the same time, the inner loop trains a model $\text{GNN}_\theta$ on condensed graph data $G^{\mathcal{S}}$ by minimizing the training loss $\ell$. Besides, to ensure the robust generalization of the condensed graph data $G^{\mathcal{S}}$ on parameter initialization distribution $P_{\theta_0}$, multiple initialization of the model parameters is required. As a result, the solving algorithm of the bi-level graph condensation actually has three nested loops, leading to huge computational costs in terms of time and GPU resources.

### 3.3 Fast Graph Condensation via Kernel Ridge Regression

To tackle the substantial computational issues, we propose to replace the $\text{GNN}_\theta$ in Equation (1) by the KRR paradigm. More specifically, KRR [6, 30] entails a convex optimization process with a closed-form solution, bypassing the resource-intensive

$\text{GNN}_\theta$ training. Consequently, the bi-level optimization in the general graph condensation framework can be simplified into a more computational efficient single-level paradigm. Moreover, KRR leverages the condensed graph data $G^{\mathcal{S}}$ as the model parameter, eliminating the need for multiple initialization to ensure robust generalization of the condensed graph data. These distinctive characteristics contribute to a significant improvement in the overall efficiency of the graph condensation. Denote $f_{G^{\mathcal{S}}}$ as the KRR model constructed by condensed data $G^{\mathcal{S}}$. Mathematically, the KRR model can be written as:

$$f_{G^{\mathcal{S}}}(G^{\mathcal{T}}) = \mathcal{K}_{\mathcal{T}\mathcal{S}}(\mathcal{K}_{\mathcal{S}\mathcal{S}} + \lambda I)^{-1} Y^{\mathcal{S}}, \tag{2}$$

where $\mathcal{K}_{\mathcal{T}\mathcal{S}} : G^{\mathcal{T}} \times G^{\mathcal{S}} \to \mathbb{R}^{N \times M}$ is a node-level kernel function. $\lambda > 0$ is the hyperparameter on regularization for preventing overfitting to mislabeled data [16]. Equation (2) leverage condensed graph data $G^{\mathcal{S}}$ to construct the KRR model and giving predictions of target graph data $G^{\mathcal{T}}$. To evaluate the predicting performance, the Mean Square Error (MSE) loss is adopted:

$$\mathcal{L}(f_{G^{\mathcal{S}}}, G^{\mathcal{T}}) = \frac{1}{2} \|Y^{\mathcal{T}} - f_{G^{\mathcal{S}}}(G^{\mathcal{T}})\|_F^2. \tag{3}$$

Then $G^{\mathcal{S}}$ is updated based on optimization algorithm `opt-alg` and the condensation loss $\mathcal{L}$:

$$G^{\mathcal{S}} = \text{opt-alg}_{G^{\mathcal{S}}}[\mathcal{L}(f_{G^{\mathcal{S}}}, G^{\mathcal{T}})]. \tag{4}$$

Iteratively conducting equations from (2) to (4) until convergence, we will get the well condensed graph data $G^{\mathcal{S}}$. Finally, the proposed graph condensation framework with KRR can be modeled as the following minimization problem:

$$\min_{G^{\mathcal{S}}} \mathcal{L}(f_{G^{\mathcal{S}}}, G^{\mathcal{T}})$$

$$s.t. \ f_{G^{\mathcal{S}}} = \mathcal{K}_{\cdot \mathcal{S}}(\mathcal{K}_{\mathcal{S}\mathcal{S}} + \lambda I)^{-1} Y^{\mathcal{S}}. \tag{5}$$

The proposed graph condensation framework is illustrated in the Figure 2(b), where incorporating KRR in graph condensation allows the bi-level optimization architecture streamlined into a single-level one. Compared with $\text{GNN}_\theta$ in Equation (1), on one hand, the convex nature of KRR allows the optimal model to be trained without iteratively training. On the other hand, KRR does not require multiple initialization of model parameters. The two

aspects contribute to the noteworthy improvement of the graph condensation efficiency.

## 3.4 Structure-based Neural Tangent Kernel (SNTK)

The role of the kernel function $\mathcal{K}$ in KRR is mapping the data to a higher-dimensional space in order to capture more intricate nonlinear structures present in the data. Different kernel functions are used to capture patterns in different types of data. Therefore, the choice of the kernel function directly affects the performance of KRR predictions and, consequently, the quality of the condensed graph data.

In recent years, Neural Tangent Kernel (NTK), which serves as a bridge between deep neural networks and kernel methods, has obtained extensive attention along with theoretical studies across enormous successful applications [13, 17, 34]. Specifically, NTK approximates behaviors of infinite-width neural networks, demonstrating the tremendous capabilities in modeling highly complex relationships among instances [13]. More importantly, harnessing the power of estimated NTK into KRR can be considered as training infinitely-wide deep neural networks in an infinite SGD training steps [1, 17], leading to efficient training without sacrificing model's expressivity. Given these advantages, the NTK provides a great opportunity to improve graph condensation that is typically based on the bi-level optimization method.

A straightforward approach using NTK [17] in graph data is to compute the kernel matrix between nodes solely based on *their features X*. For two nodes $v_i$ and $v_j$ with features $x_i$ and $x_j$, respectively, denote $\Theta_{ij}^{(0)} = \Sigma_{ij}^{(0)} = x_i \cdot x_j$ as initialization. The recursive NTK workflow is given by:

$$\Theta_{ij}^{(L)} = \Theta_{ij}^{(L-1)} \dot{\Sigma}_{ij}^{(L)} + \Sigma_{ij}^{(L)}, \tag{6}$$

in which

$$\Sigma_{ij}^{(L)} = \alpha \mathbb{E}_{(a,b) \sim \mathcal{N}(0, \Lambda_{ij}^{(L)})} [\sigma(a)\sigma(b)], \tag{7}$$

$$\dot{\Sigma}_{ij}^{(L)} = \alpha \mathbb{E}_{(a,b) \sim \mathcal{N}(0, \Lambda_{ij}^{(L)})} [\dot{\sigma}(a)\dot{\sigma}(b)], \tag{8}$$

$$\Lambda_{ij}^{(L)} = \begin{bmatrix} \Sigma_{ii}^{(L-1)} & \Sigma_{ij}^{(L-1)} \\ \Sigma_{ji}^{(L-1)} & \Sigma_{jj}^{(L-1)} \end{bmatrix}, \tag{9}$$

where $\dot{\sigma}(a)$ is the derivative with respect to activation $\sigma(a)$. $\alpha$ is the coefficient related to activation $\sigma$. $\Theta_{ij}^{(L)}$ is the kernel value after $L$ iterations.

However, calculating the kernel value only based on node features might easily lead to low-quality condensed graphs. Because the structural information in graphs can provide crucial insights into the relationships, dependencies, and interactions among nodes. By disregarding structural information, important contextual information and patterns in the graphs may be overlooked. Hence, considering the structural information in NTK for graph condensation is essential.

To tackle this issue, we introduce a novel kernel method, namely Structure-based Neural Tangent Kernel (SNTK), to capture the structure of graphs by connecting local neighborhood aggregation and the NTK method. Based on the homophily assumption [26], SNTK involves integrating information from neighboring nodes into enhancing node representation learning, aiming to capture nodes' context and relationships. More formally, the node neighborhood aggregation of SNTK can be defined as follows:

$$h_i = c_i \sum_{p \in \mathcal{N}\{i\} \cup \{i\}} x_p, \tag{10}$$

where $\mathcal{N}\{i\} \cup \{i\}$ indicates the set consisting of node $v_i$ and its neighbors $\mathcal{N}\{i\}$. To avoid imbalanced information propagation, the aggregation coefficient is set to $c_i = (\| \sum_{p \in \mathcal{N}\{i\} \cup \{i\}} x_p \|_2)^{-1}$. With local neighborhood aggregation, the SNTK kernel can be formulated as:

$$\hat{\Sigma}_{ij}^{(0)} = h_i \cdot h_j = c_i c_j \sum_{p \in \mathcal{N}\{i\} \cup \{i\}} \sum_{q \in \mathcal{N}\{j\} \cup \{j\}} \Sigma_{pq}^{(0)}, \tag{11}$$

$$\hat{\Theta}_{ij}^{(0)} = h_i \cdot h_j = c_i c_j \sum_{p \in \mathcal{N}\{i\} \cup \{i\}} \sum_{q \in \mathcal{N}\{j\} \cup \{j\}} \Theta_{pq}^{(0)}, \tag{12}$$

$$\hat{\Theta}_{ij}^{(L)} = \hat{\Theta}_{ij}^{(L-1)} \dot{\Sigma}_{ij}^{(L)} + \hat{\Sigma}_{ij}^{(L)}. \tag{13}$$

Real-world scenarios often necessitate $K$ ($K > 1$) rounds of neighborhood aggregation to capture information from nodes' $K$-hop neighbors. Therefore, the aggregation followed by $L$ iterations of kernel matrix will be recursively iterated $K$ times.

## 3.5 Matrix Form of Structure-based Neural Tangent Kernel (SNTK)

To enable efficient acceleration computations on GPUs, we formulate the SNTK computational process in the matrix format. Given two different graphs $G = \{X, A, Y\}$ and $G' = \{X', A', Y'\}$, the initialization of SNTK is set as: $\Theta^{(0)} = \Sigma^{(0)} = X(X')^\top$. Denote the matrix form aggregation as $\hat{\Sigma} = Aggr(\Sigma)$, the matrix form for Equation (11) and (12) is given by:

$$\hat{\Sigma}^{(0)} = \hat{A}(C \odot \Sigma^{(0)})(\hat{A}')^\top, \tag{14}$$

$$\hat{\Theta}^{(0)} = \hat{A}(C \odot \Theta^{(0)})(\hat{A}')^\top, \tag{15}$$

where $\hat{A}$ and $\hat{A}'$ are the self-looped adjacency matrix of graphs $G$ and $G'$. $C$ is the aggregation coefficient matrix, where $C_{ij} = c_i c_j$. $(\cdot)^\top$ indicates the transpose operation, $\odot$ is the element-wise product. Then according to [1], the matrix form of the recursive iteration in Equation (13) is:

$$\dot{\Sigma}^{(L)} = \frac{1}{2\pi} \left[ \pi - arccos(\hat{\Sigma}^{(L-1)}) \right], \tag{16}$$

$$\hat{\Sigma}^{(L)} = \frac{1}{2\pi} \left[ \pi - arccos(\hat{\Sigma}^{(L-1)}) + \sqrt{1 - (\hat{\Sigma}^{(L-1)})^2} \right], \tag{17}$$

$$\hat{\Theta}^{(L)} = \hat{\Theta}^{(L-1)} \odot \dot{\Sigma}^{(L)} + \hat{\Sigma}^{(L)}, \tag{18}$$

where $arccos(\hat{\Sigma}^{(L-1)})$ and $(\hat{\Sigma}^{(L-1)})^2$ indicate element-wise arc-cosine and squaring operation. To sum up, for $K$ times neighborhood aggregation, where each aggregation followed by $L$ iterations of Equation (18), the workflow for the SNTK is illustrated in **Algorithm 1**.

Finally, the algorithm for proposed GC-SNTK, which incorporates KRR paradigm and SNTK, is summarised in **Algorithm 2**.

**Algorithm 1** Structure-based Neural Tangent Kernel (SNTK)

1: **Input:** Graphs $G = \{X, A, Y\}$, $G' = \{X', A', Y'\}$.
2: **Output:** Kernel matrix $\hat{\Theta}_{(K)}^{(L)}$.
3: **Initialize:** $\Theta_{(0)}^{(0)} = \Sigma_{(0)}^{(0)} = X(X')^{\top}$, $K$, $L$.
4: **for** $k = 1, 2, ..., K$ **do**
5:    **if** $k = 1$ **then**
6:       $\hat{\Sigma}_{(k)}^{(0)} = Aggr(\Sigma_{(0)}^{(0)})$, $\hat{\Theta}_{(k)}^{(0)} = Aggr(\Theta_{(0)}^{(0)})$.
7:    **else**
8:       $\hat{\Sigma}_{(k)}^{(0)} = Aggr(\hat{\Sigma}_{(k-1)}^{(L)})$, $\hat{\Theta}_{(k)}^{(0)} = Aggr(\hat{\Theta}_{(k-1)}^{(L)})$.
9:    **end if**
10:    **for** $l = 1, 2, ..., L$ **do**
11:       Update $\dot{\Sigma}_{(k)}^{(l)} \leftarrow \hat{\Sigma}_{(k)}^{(l-1)}$; $\Sigma_{(k)}^{(l)} \leftarrow \hat{\Sigma}_{(k)}^{(l-1)}$; $\hat{\Theta}_{(k)}^{(l)} \leftarrow \hat{\Theta}_{(k)}^{(l-1)}$,
      $\dot{\Sigma}_{(k)}^{(l)}$, $\hat{\Sigma}_{(k)}^{(l)}$ by Equation (16) - (18).
12:    **end for**
13: **end for**

---

**Algorithm 2** GC-SNTK

1: **Input:** Target graph data $G^{\mathcal{T}}$ with $N$ nodes.
2: **Output:** Condensed graph data $G^{\mathcal{S}}$ with $M(M \ll N)$ nodes.
3: **Initialize:** Random initialize $G^{\mathcal{S}}$.
4: **while** not converge **do**
5:    Calculate $\mathcal{K}_{\mathcal{TS}}$ and $\mathcal{K}_{\mathcal{SS}}$ by **Algorithm** 1.
6:    Calculate loss function $\mathcal{L}(f_{G^S}, G^{\mathcal{T}})$ by Equation (3):

$$\mathcal{L}(f_{G^S}, G^{\mathcal{T}}) = \frac{1}{2}||Y^{\mathcal{T}} - f_{G^S}(G^{\mathcal{T}})||_F^2.$$

7:    Update $G^{\mathcal{S}}$ by Equation (4):

$$G^{\mathcal{S}} = \texttt{opt-alg}_{G^S}\left[\mathcal{L}(f_{G^S}, G^{\mathcal{T}})\right].$$

8: **end while**

---

## 3.6 Computational Complexity Analysis

To theoretically demonstrate that the computational complexity of our proposed GC-SNTK method is lower than that of bi-level method, i.e., GCond, we conduct a comprehensive computational complexity analysis in this part. The notations used in this part are as follows: $N$ and $M$ represents the number of nodes in the original dataset and the condensed data, $d$ stands for the node feature dimensionality, $w$ signifies the GCN hidden layer width, and $k$ represents the average number of neighbors per node (in the GCond method, $k = M$ since the adjacency relationship between condensed nodes is constructed as a fully connected weighted graph). Additionally, $t_{in}$ and $t_{out}$ denote the number of iterations in the inner loop and outer loop, respectively, and $R$ corresponds to the number of training epochs, which is the initialization number of model parameters in GCond.

The computational complexity of GCond method during the inner loop of GCN training is $O(t_{in}(M^2d + Mdw))$. The outer loop of GCond consists of two parts: 1) optimizing the node features $X^{\mathcal{S}}$ or optimizing the MLP model used to update $A^{\mathcal{S}}$; 2) updating $A^{\mathcal{S}}$ according to the updated node features $X^{\mathcal{S}}$. Let the number of iterations for optimizing node features $X^{\mathcal{S}}$ be $t_X$ and the number

of iterations for optimizing MLP be $t_A$ ($t_X + t_A = t_{out}$). Then the computational complexity of the outer loop is: $O(t_X(Nkd + M^2dw) + t_A(M^2dw))$. Therefore, the total computational complexity of the **GCond** method is $O(Rt_{out}t_{in}(M^2d + Mdw) + Rt_X(Nkd) + Rt_{out}(M^2dw))$.

On the other hand, the computational complexity of proposed GC-SNTK mainly consists of two parts: kernel matrix calculation and KRR model construction. For the former, in practical experiments, parameters $K$ and $L$ are usually set to be relatively small (e.g., in the ogbn-arxiv dataset, $K = L = 1$). So, these parameters linearly affect the computational complexity and that is $O(MNk^2 + MN)$. For the latter, it is $O(NM^2)$. Therefore, the total computational complexity of the **GC-SNTK** method is $O(RMNk^2 + RNM^2)$. Although the time complexity of both GCond and GC-SNTK are linearly increases with the target graph nodes number $N$, GC-SNTK proves advantageous due to its single loop and rapid convergence, leading to a faster execution speed compared to GCond.

## 4 EXPERIMENT

In this section, a performance comparison is conducted for node classification models trained on condensed data to evaluate the expressive ability of the condensed graph data. Subsequently, we conduct experiments on extremely small condensation sizes to measure the effectiveness of the condensation methods. Additionally, the efficiency of various condensation methods is assessed. The experimental results serve to validate the conclusions drawn in the computational analysis in Section 3.6. Moreover, an examination of the generalization capabilities of the condensed data is presented, along with an ablation study of the kernel method. Lastly, a sensitivity analysis of the parameters is conducted. All the experiments are conducted on an NVIDIA RTX 3090 GPU.

## 4.1 Experimental Settings

**Datasets**. Four different scale benchmark datasets in the graph domain are adopted, including Cora, Pubmed [43], Ogbn-arxiv [15], and Flickr [44]. These datasets vary in size, ranging from thousands of nodes to hundreds of thousands of nodes. Additionally, these datasets are categorized into two different settings: transductive (Cora, Pubmed, and Ogbn-arxiv) and inductive (Flickr). The details of the datasets are shown in Table 1. To ensure the transductive setting during experiments, we apply the graph convolution on the entire graph data of Cora, Pubmed, and Ogbn-arxiv datasets to facilitate information propagation between different nodes. Graph convolution can be modeled as $\tilde{X} = \tilde{A}X$, where $X$ is the node feature matrix. $\tilde{A} = \tilde{D}^{-\frac{1}{2}}\hat{A}\tilde{D}^{-\frac{1}{2}}$, $\hat{A} = A+I$, $A$ is the adjacency matrix. $\tilde{D} = diag(\sum_j \hat{A}_{1j}, \sum_j \hat{A}_{2j}, ..., \sum_j \hat{A}_{nj})$.

**Baselines**. To evaluate the effectiveness of GC-SNTK, we compare it against various graph condensation methods serving as baselines, including three core-set methods Random, Herding [39], and K-center [33], as well as a simplified GCond method named as One-Step [18]. Additionally, we consider two different versions of the previous state-of-the-art (SOTA) graph condensation method, denoted as GCond (X) and GCond (X, A) [19]. The former condense the graph data into the version only including node features, while the latter including both node features and structural information. The GCond framework offers flexibility in utilizing different GNNs

**Table 1: Details of the datasets.**

| Dataset | | Transductive | | Inductive |
|---|---|---|---|---|
| | Cora | Pubmed | Ogbn-arxiv | Flickr |
| #Nodes | 2,708 | 19,717 | 169,343 | 89,250 |
| #Edges | 5,429 | 44,338 | 1,166,243 | 899,756 |
| #Classes | 7 | 3 | 40 | 7 |
| #Features | 1,433 | 500 | 128 | 500 |
| Split Train | 140 | 60 | 90,941 | 44,625 |
| Val. | 500 | 500 | 29,799 | 22,312 |
| Test | 1,000 | 1,000 | 48,603 | 22,313 |

for the condensation and testing stages. Hence, there are various possible combinations available for this approach. For the sake of generality, in the experimental section, unless otherwise specified, we default to using the best-performing combination of SGC-GCN for all GCond methods. Specifically, the SGC [41] is employed for condensation, whereas the GCN [20] is utilized for testing.

**Parameter Settings**. For SNTK, we tune the number of $K$ and $L$ in a range of $\{1, 2, 3, 4, 5\}$. The hyper-parameter $\lambda$ of KRR is tuned within the range from $10^{-6}$ to $10^{6}$. The values for learning rate are $\{0.1, 0.05, 0.01, 0.005, 0.001, 0.0005, 0.0001\}$. The experimental parameter settings for GC-SNTK is given in Table 2. Regarding the GCond and One-Step methods, we follow the settings described in the original papers [18, 19].

## 4.2 Performance Comparison of Condensed Graph Data

This part aims to assess the representation capability of the condensed graph data in three different condensation scales. We train nodes classification models on the condensed data to predict the labels of the test set. The classification accuracy serves as the measure for evaluating the representation capability of condensed graph data. For each dataset, we opt for three distinct condensation ratios. In the transductive scenario, $Ratio = \frac{M}{N}$. In the inductive scenario, $Ratio = \frac{M}{|training\ set|}$.

As shown in Table 3, the proposed GC-SNTK method demonstrates its remarkable graph condensation capability. Even at condensation ratios of 0.1% (Flickr) and 0.05% (Ogbn-arxiv), GC-SNTK achieves **99%** and **89%** of the performance of the GCN model trained on the full training set, respectively. Moreover, GC-SNTK even outperforms the GCN model on Cora and Pubmed datasets. Specifically, with condensation ratios of 1.3% (Cora) and 0.08% (Pubmed), GC-SNTK achieves performance levels of **101%** and **102%** compared to that of the GCN model trained on the full training data.

The results on Cora and Pubmed datasets show that GC-SNTK can improve efficiency, eliminate redundancies, and retain the most representative information in the original data. In other words, the proposed method can reduce datasets scale and enhance the entities' representation capability without scarifying the predictive performance.

**Table 2: Details of the parameters setting in GC-SNTK.**

| Dataset | Learning Rate | Ridge ($\lambda$) | $K$ | $L$ |
|---|---|---|---|---|
| Cora | 0.01 | 1 | 2 | 2 |
| Pubmed | 0.01 | 0.001 | 2 | 2 |
| Ogbn-arxiv | 0.001 | 0.00001 | 1 | 1 |
| Flickr | 0.001 | 0.00001 | 1 | 1 |

## 4.3 Performance with Extremely Small Condensation Size

This part explores the variation in the representational performance of condensed graph data as the condensation scale decreases. We continuously reduce the number of nodes in the condensed data and observe how their performance change. The experimental results are presented in Figure 3. Due to the limitations of the GCond algorithm, it is unable to condense graph data into node counts smaller than the number of node categories. Therefore, when the number of nodes in the condensed data is smaller than the number of classes in the target dataset (Figure 3(a) - 3(d)), only GC-SNTK is involved in the experiments.

The proposed GC-SNTK method maintains strong performance even the scale of synthetic graphs is extremely small. Predictive performance of the model trained on condensed data only significantly drops when the number of nodes in the condensed data is smaller than the number of node categories. Additionally, on the Ogbn-arxiv dataset, our method achieves 63% accuracy (Figure 3(c)) even when the condensed data contains only 20 nodes (with 40 categories).

The experimental results in Figure 3(e) - 3(h) show the performance comparison of GC-SNTK, with GCond and One-Step methods at smaller condensation scales. Overall, GC-SNTK exhibits outstanding performance across the majority of condensation scales on all four datasets. Particularly on the Ogbn-arxiv dataset, GC-SNTK is significantly better than the other two methods, achieving a higher accuracy of **3%**. In contrast, GCond performs slightly worse than GC-SNTK, while the simplified optimization process of the One-Step method leads to the poorest performance.

## 4.4 Condensation Efficiency

This part focuses on evaluating the time efficiency of GC-SNTK and One-Step (faster than GCond) from an empirical perspective. As the optimization of condensed data progresses, the performance of node classification models trained on compressed data also changes over time. Therefore, we evaluate GC-SNTK, in terms of its time efficiency by observing the correlation between model performance and the time spent on optimizing condensed data. This correlation is illustrated by the curves in Figure 4, where the $y$-axis represents model performance and the $x$-axis represents the time consumption. Each curve stops when it reaches the corresponding performance indicated in Table 3.

As shown in Figure 4, GC-SNTK exhibits remarkable time efficiency across all four datasets. For instance, GC-SNTK can condense the Cora and Pubmed datasets to 70 and 30 nodes in 4.01 and 3.37 seconds, which is significantly (**61.5** and **23.6** times) faster than One-Step method (Cora: 246.7s, Pubmed: 79.6s), and achieving a higher accuracy performance. Moreover, our method maintains advantages on large datasets like Ogbn-arxiv and Flickr. By condensing these two datasets to 90 and 44 nodes, GC-SNTK (Ogbn-arxiv: 871.5s, Flickr: 163.0s) outperforms the One-Step method (Ogbn-arxiv: 3161.1s, Flickr: 441.3s) by **3.6** and **2.7** times faster, respectively.

Compared to One-Step, a bi-level optimization approach simplified to improve the efficiency of condensation by performing each loop only once, GC-SNTK both ensures its performance of

Table 3: Performance evaluation of the condensed data. The last column displays the classification accuracy obtained by the Graph Convolutional Neural Network (GCN) model trained on the complete training set.

| Dataset | Ratio (Size) | Random | Herding | K-Center | One-Step | Gcond | | GC-SNTK (Our) | | Full (GCN) |
|---|---|---|---|---|---|---|---|---|---|---|
| | | | | | | X | X, A | X | X, A | |
| Cora | 1.30% (35) | 63.6±3.7 | 67.0±1.3 | 64.0±2.3 | 80.2±0.73 | 75.9±1.2 | 81.2±0.7 | **82.2±0.3** | 81.7±0.7 | |
| | 2.60% (70) | 72.8±1.1 | 73.4±1.0 | 73.2±1.2 | 80.4±1.77 | 75.7±0.9 | 81.0±0.6 | **82.4±0.5** | 81.5±0.7 | 81.1±0.5 |
| | 5.20% (140) | 76.8±0.1 | 76.8±0.1 | 76.7±0.1 | 79.8±0.64 | 76.0±0.9 | 81.1±0.5 | **82.1±0.1** | 81.3±0.2 | |
| Pubmed | 0.08% (15) | 69.5±0.5 | 73.0±0.7 | 69.0±0.6 | 77.7±0.12 | 59.4±0.7 | 78.3±0.2 | **78.9±0.7** | 71.8±6.8 | |
| | 0.15% (30) | 73.8±0.8 | 75.4±0.7 | 73.7±0.8 | 77.8±0.17 | 51.7±0.4 | 77.1±0.3 | **79.3±0.3** | 74.0±4.9 | 77.1±0.3 |
| | 0.30% (60) | 77.9±0.4 | 77.9±0.4 | 77.8±0.5 | 77.1±0.44 | 60.8±1.7 | 78.4±0.3 | **79.4±0.3** | 76.4±2.8 | |
| Ogbn-arxiv | 0.05% (90) | 47.1±3.9 | 52.4±1.8 | 47.2±3.0 | 59.2±0.03 | 61.3±0.5 | 59.2±1.1 | 63.9±0.3 | **64.4±0.2** | |
| | 0.25% (454) | 57.3±1.1 | 58.6±1.2 | 56.8±0.8 | 60.1±0.75 | 64.2±0.4 | 63.2±0.3 | **65.5±0.1** | 65.1±0.8 | 71.4±0.1 |
| | 0.50% (909) | 60.0±0.9 | 60.4±0.8 | 60.3±0.4 | 60.0±0.12 | 63.1±0.5 | 64.0±0.4 | **65.7±0.4** | 65.4±0.5 | |
| Flickr | 0.10% (44) | 41.8±2.0 | 42.5±1.8 | 42.0±0.7 | 45.8±0.47 | 45.9±0.1 | 46.5±0.4 | 46.6±0.3 | **46.7±0.1** | |
| | 0.50% (223) | 44.0±0.4 | 43.9±0.9 | 43.2±0.1 | 46.6±0.13 | 45.0±0.2 | **47.1±0.1** | 46.7±0.1 | 46.8±0.1 | 47.2±0.1 |
| | 1.00% (446) | 44.6±0.2 | 44.4±0.6 | 44.1±0.4 | 45.4±0.31 | 45.0±0.1 | **47.1±0.1** | 46.6±0.2 | 46.5±0.2 | |

(a) Cora    (b) Pubmed    (c) Ogbn-arxiv    (d) Flickr

(e) Cora    (f) Pubmed    (g) Ogbn-arxiv    (h) Flickr

Figure 3: The comparison of node classification accuracy. (a)-(d) illustrate the performance variation of GC-SNTK as the condensation size decreases to a single node. (e)-(f) represent the performance comparison of GC-SNTK, GCond, and One-Step methods at extremely small condensation sizes.

condensed graph data and significantly speeds up the condensation process.

## 4.5 Generalization of Condensed Data

This part investigates the generalization of the condensed data across different graph node classification models. Specifically, we use the condensed data to train various models, including GCN [20, 32, 45], SGC [4, 41], APPNP [12, 27], GraphSAGE (SAGE) [14], and KRR [36, 40, 47] to assess their generalization performance. For the GCond method, we use four different GNN models (GCN, SGC, APPNP, SAGE) in the condensation process and evaluate the condensed graph data on all five models. The experimental results are presented in Table 4.

Even though GC-SNTK is a non-neural network model, the data condensed by it still performs well when training neural network models. For instance, the average performance of the prediction models trained on the condensed data by GC-SNTK from the Cora and Pubmed datasets achieve accuracies of 78.0%

and 75.7%, respectively. The GCond method can achieve better data condensation quality and generalization performance when using SGC as the condensation model. In contrast, the performance of the condensed data on the KRR model is poor. As a result, the overall performance of the GCond is inferior to the GC-SNTK method.

## 4.6 Ablation Study

In this section, we choose widely used dot product kernel function, as well as NTK for comparison, to test the impact of different kernel functions on the quality of condensed graph data. Experimental results demonstrate that our proposed structure-based neural tangent kernel method can better capture the information of graph data and improve the quality of condensed graph data.

As shown in Table 5, the utilization of SNTK exhibits a more significant impact on the results across all datasets. SNTK surpasses the performance of the other two kernel functions, thereby providing evidence of its influential role in measuring the similarities

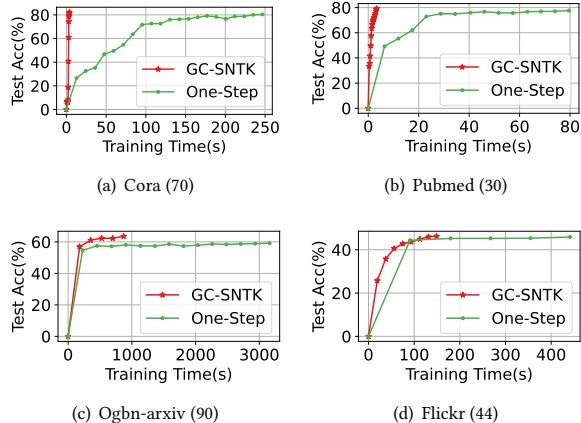

(a) Cora (70)  (b) Pubmed (30)

(c) Ogbn-arxiv (90)  (d) Flickr (44)

**Figure 4: Condensation efficiency consumption on the four datasets (the number after the name of dataset is the nodes size of the condensed data).**

**Table 4: The generalization capacity of the condensed data. We utilise various models to condense (C) graph data and test (T) the performance of models trained on the condensed data.**

| Dataset | C/T | GCN | SGC | APPNP | SAGE | KRR | Avg. |
|---|---|---|---|---|---|---|---|
| Cora (70) | **GCN** | 70.6 | 68.7 | 69.8 | 60.2 | 38.9 | 61.6 |
| | SGC | 80.1 | 79.3 | 78.5 | 78.2 | 73.0 | 77.8 |
| | APPNP | 73.5 | 73.1 | 72.1 | 72.3 | 63.7 | 70.9 |
| | SAGE | 77.0 | 77.7 | 77.1 | 76.1 | 23.7 | 66.3 |
| | **GC-SNTK** | 78.0 | 76.9 | 75.7 | 77.0 | 82.4 | **78.0** |
| Pubmed (30) | GCN | 50.4 | 50.5 | 54.8 | 51.3 | 31.2 | 47.6 |
| | SGC | 77.1 | 76.0 | 77.1 | 77.1 | 45.2 | 70.5 |
| | APPNP | 68.0 | 60.7 | 77.5 | 73.7 | 65.7 | 69.1 |
| | SAGE | 51.9 | 58.0 | 69.9 | 65.3 | 41.3 | 57.3 |
| | **GC-SNTK** | 76.1 | 73.2 | 75.6 | 74.1 | 79.3 | **75.7** |

**Table 5: Ablation study of the kernels on different datasets.**

| Dataset | Kernels | | |
|---|---|---|---|
| | Dot Product | NTK | SNTK |
| Cora (70) | 78.9±0.3 | 80.9±1.4 | **82.4±0.5** |
| Pubmed (30) | 77.4±1.3 | 78.8±0.5 | **79.3±0.3** |
| Ogbn-arxiv (90) | 63.8±0.1 | 63.9±0.1 | **64.4±0.2** |
| Flickr (44) | 41.9±0.1 | 43.1±1.8 | **46.6±0.3** |

among graph nodes and improving the quality of condensed graph data.

### 4.7 Parameter Sensitivity Analysis

In GC-SNTK, there are three parameters that influence the model performance: ridge $\lambda$ in KRR, aggregation times $K$, and iteration count $L$ in SNTK. Therefore, this part experimentally studies the impact of these three parameters on graph condensation. The results of the experiment indicate that our method is not sensitive to these parameters, and as long as they are within a reasonable range, the performance differences exhibited by the model are not significant.

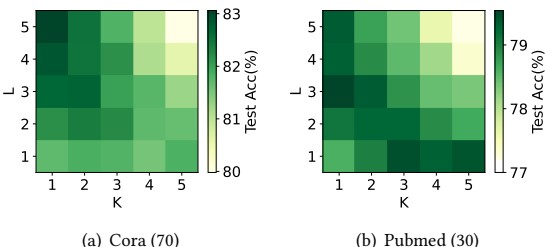

(a) Cora (70)  (b) Pubmed (30)

**Figure 5: Sensitivity analysis of $K$ and $L$ in SNTK. We choose different values of $K$ and $L$ to pose different SNTKs. Their performances are represented as different colors shown above.**

As depicted in Fig. 5, we investigate the impacts of different values of $K$ and $L$. We change these variations to create 25 distinct kernel functions, which are subsequently applied to the proposed graph condensation framework. The classification accuracy on the test set is visualized using a color scale, where darker colors indicate higher accuracy. Surprisingly, even among these 25 kernel functions, their accuracy only differed by 3.1% on the Cora dataset, and 2.3% on the Pubmed dataset. Hence, it is evident that GC-SNTK is not sensitive to the parameters of SNTK.

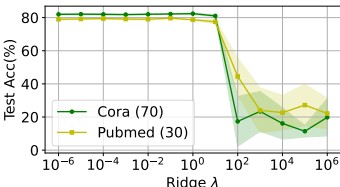

**Figure 6: Sensitivity analysis of the ridge $\lambda$ in Eq. (2). We choose 13 different ridge values from $1 \times 10^{-6}$ to $10^6$ to observe the impact of ridge on results.**

For the ridge $\lambda$ of KRR, we conducted experiments on the Cora and Pubmed datasets, exploring 13 different magnitudes ranging from $1 \times 10^{-6}$ to $10^6$. The experimental results, as shown in Fig. 6, demonstrate that GC-SNTK is also not sensitive to the ridge $\lambda$. As long as the ridge is less than or equal to 1, the experimental results do not differ significantly. Therefore, we can conclude that the proposed method GC-SNTK is not sensitive to these parameters.

### 5 CONCLUSION

This paper proposes a novel dataset condensation framework (GC-SNTK) for condensing graph-structured data. Specifically, GC-SNTK transforms the bi-level condensation problem into a single-level optimization task as the KRR paradigm. In addition, a structured-based kernel function (SNTK) is introduced to enhance the quality of the condensed data in KRR. Based on NTK and neighborhood aggregation, SNTK can simultaneously leverage the node features and structural information of graphs to capture the complex dependencies among nodes. The experimental results demonstrate the superiority of our proposed GC-SNTK method in efficiency and efficacy. Furthermore, the proposed GC-SNTK performs promising generalization capability across various GNN architectures on graph-structured data.

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
