# OpenReview forum: "Fast Graph Condensation with Structure-based Neural Tangent Kernel"
_ACM.org/TheWebConf/2024/Conference — TheWebConf24_

### Official Review · Reviewer_W5kS · 2023-11-13

**Novelty:** 3
**Technical Quality:** 4

**Review:**

In this paper, the authors advocate a paradigm shift for addressing the graph condensation problem. Instead of resorting to the iterative training of Graph Neural Networks (GNNs) within the inner loop of bi-level optimization, they propose a novel approach. Specifically, their proposed method entails reformulating the problem as a Kernel Ridge Regression (KRR) task.

positive:
1.  propose a principle way based on kernel ridge regres-sion to significantly improve graph condensation efficiency, which can be achieved by replacing the inner GNNs training (i.e., inner loop) via Structure-based Neural Tangent Kernel in the KRR paradigm.
2. The performance is good, comparing with baseline.


negative:

1. My main concern about this paper is its novelty. The core contribution of this paper, the proposed SNTK, is quite similar to GNTK [1]. The difference is that GNTK has an extra pooling operator because GNTK is to compute graph-graph similarity. Thus, the novelty of this paper is relatively low. In addition, most of the content from Section 3.4 is included in the GNTK paper [1], and the so-called matrix from SNTK (Section 3.5) is a straightforward extension of Section 3.4. In fact, in the official implementation of GNTK (e.g., line 112 in [2]), they use a Kronecker product of the adjacency matrices to present the aggregation operations, which can be easily transformed into the Eqs. (14) and (15) via the common property of Kronecker product.

[1] Du, Simon S., Kangcheng Hou, Russ R. Salakhutdinov, Barnabas Poczos, Ruosong Wang, and Keyulu Xu. "Graph neural tangent kernel: Fusing graph neural networks with graph kernels." Advances in neural information processing systems 32 (2019).

[2] https://github.com/KangchengHou/gntk/blob/master/gntk.py

[3] Wu, Jun, Lisa Ainsworth, Andrew Leakey, Haixun Wang, and Jingrui He. "Graph-Structured Gaussian Processes for Transferable Graph Learning." In Thirty-seventh Conference on Neural Information Processing Systems. 2023.

2. The code of this work is not provided, it reproducibility is unlcear.

3. This work is more related to Graph Track, than Web Mining track.

**Questions:**

see weakness.

**Reviewer Confidence:**

4: The reviewer is certain that the evaluation is correct and very familiar with the relevant literature

**Scope:**

1: The work is irrelevant to the Web

---

### Official Review · Reviewer_a3km · 2023-11-22

**Novelty:** 6
**Technical Quality:** 6

**Review:**

The paper proposes a novel approach, GC-SNTK, to address the computational resource costs associated withGNNs when handling large-scale graph-structured data. The authors compare against the computationally intensive bi-level optimization architecture used in existing methods and suggest reformulating the graph condensation problem as a Kernel Ridge Regression task. The proposed approach involves a method to capture graph topology, improving efficiency without sacrificing predictive performance.

Pros:
- The paper addresses a notable challenge in GNNs by proposing a principled way to improve efficiency through kernel ridge regression and moreover, the introduction of the GC-SNTK framework is a notable contribution, demonstrating effectiveness and efficiency in synthesizing smaller yet information-rich graphs for various GNN architectures while maintaining generalization performance. I think that the demonstrated generalization capability across different GNN architectures is a strong point.


Cons-
- While the paper introduces the GC-SNTK framework as a novel approach, a more direct comparison with existing bi-level optimization-based methods, highlighting advantages and limitations, would provide a clearer context for the proposed solution.

**Questions:**

NA

**Reviewer Confidence:**

3: The reviewer is confident but not certain that the evaluation is correct

**Scope:**

4: The work is relevant to the Web and to the track, and is of broad interest to the community

---

### Official Review · Reviewer_pfpi · 2023-11-23

**Novelty:** 6
**Technical Quality:** 6

**Review:**

## Updates after Authors' Response

I have read the response from the authors, which have sufficiently addressed my questions in the original review. I keep my scores in the original review.

---
## Summary

This work proposed a novel approach for graph condensation called GC-SNTK. Specifically, it formulates the problem as a Kernel Ridge Regression (KRR) task using a Structure-based Neural Tangent Kernel (SNTK). As a result, it removes the necessity of employing complex three nested loops for bi-level GNN optimization and parameter initialization as in existing approaches. The proposed approach is clearly described with equations, algorithms blocks and intuitive illustrations. The authors conducted comprehensive experiments, where they show the proposed approach is up to 61.5x faster than the state-of-the-art baseline while achieving comparable or better performance.

## Strengths

- The problem motivation is very clear; the well-designed figures and educative discussions on the related works greatly help the readers to understand the background and motivation.
- The idea of replacing the complex nested loops in existing approaches with Kernel Ridge Regression (KRR) and Structure-based Neural Tangent Kernel (SNTK) is novel,  interesting, and well-supported by existing theories.
- The proposed approach is described in a clear and easy-to-adapt way with well-explained equations, algorithms blocks and intuitive illustrations.
- The experiments results demonstrate clear advantage in efficiency of the proposed approach to the existing baselines, while maintaining similar or better performance as state-of-the-art approaches.

## Weakness

- It is a bit strange to include KRR along with four other GNN approaches in average performance calculation in Section 4.5 (Generalization of Condensed Data). As the results show that that graphs condensed with bi-level GNN training does not perform well on KRR model, it seems not fair to include KRR in the average here if the condensed graph is expected to be consumed in GNN models in the downstream. If non-GNN downstream models are also expected, then (1) more non-GNN approaches should be considered, and (2) the average performance for GNN and non-GNN approaches should be separately reported.
- While GCond results in Table 3 is based on bi-level training on GCN, Table 4 shows that  bi-level optimization on SGC helps improve the performance on GCond. The authors should consider adding these results (from a stronger baseline) to Table 3 as well.

## Other Suggestions

- In Table 4, the four models other than GC-SNTK in Table 4 should be noted (e.g., with prefix) that the condensation is done by GCond.

**Questions:**

Please address the weakness section of the review.

**Reviewer Confidence:**

4: The reviewer is certain that the evaluation is correct and very familiar with the relevant literature

**Scope:**

4: The work is relevant to the Web and to the track, and is of broad interest to the community

---

### Official Review · Reviewer_pQnc · 2023-11-24

**Novelty:** 6
**Technical Quality:** 6

**Review:**

**Summary**

This paper proposed a novel framework of Graph Condensation using Neural Tangent Kernel (NTK) instead of previous bi-level methods. It points out the problems that previous methods suffer from, then it reforms the graph condensation as a Kernel Ridge Regression (KRR) task and designs a structure-based NTK to deal with structural property. After conducting comprehensive experiments, it shows that this method can overcome the problems effectively and perform well on real-world benchmarks.

**Pros**

1. Transforming the graph condensation problem into KRR task is a very novel insight. It ingeniously tackles the inner-outer optimization process into a single optimization loop.

2. It is convincing to conduct comprehensive experiments to show the effectiveness of this method.

3. This paper theoretically analysis the computational complexity to show the lower computational cost.

4. The organization of the article is clear and logical.

**Cons**

1. The types of experimental datasets are insufficient, since the three datasets are all citation networks. It is necessary to supplement different types of experimental datasets.

2. The authors indicate that bi-level methods suffer from training instability, however, they do not verify how stability performs on their method.

**Questions:**

Q1. From my perspective, the node neighborhood aggregation (NA) of SNTK has the  similar form with the one of GCNs'. What's the motivation to design NA in this way?

Q2. For the optimization objective, since the labels of nodes are categories, why use MSE to evaluate the prediction quality instead of using CrossEntropy?

**Reviewer Confidence:**

3: The reviewer is confident but not certain that the evaluation is correct

**Scope:**

3: The work is somewhat relevant to the Web and to the track, and is of narrow interest to a sub-community

---

### Decision · Program_Chairs · 2024-01-22

**Decision:**

Accept

**Comment:**

The authors propose a graph condensation approach that builds on SNTK and uses kernel ridge regression. There were some concerns on novelty on how the method can be decomposed to GNTK, but I do not find this to be a significant issue for novelty. There were some valid concerns on stronger baselines not being compared. One reviewer noted that the work is more related to the web track, but I find that the work is still quite relevant to the Web Mining track. Overall, I find there are some valid cons, but generally the pros outweight the cons.